# Effects of Chitosan on Loading and Releasing for Doxorubicin Loaded Porous Hydroxyapatite–Gelatin Composite Microspheres

**DOI:** 10.3390/polym14204276

**Published:** 2022-10-12

**Authors:** Meng-Ying Wu, Yu-Hsin Liang, Shiow-Kang Yen

**Affiliations:** 1Department of Materials Science and Engineering, National Chung Hsing University, Taichung 40227, Taiwan; 2Department of Orthopaedic Surgery, Taichung Armed Force General Hospital, Taichung 41168, Taiwan

**Keywords:** porous hydroxyapatite/gelatin microspheres, doxorubicin, chitosan, shielding effect, drug loading and releasing

## Abstract

Porous hydroxyapatite–gelatin (Hap–Gel) composite microspheres derived by wet chemical methods were used as carriers of doxorubicin (DOX) coupled with chitosan (Chi) for treating cancers. Through X-ray diffraction, specific surface area porosimetry, chemisorption analysis and inductively coupled plasma mass spectrometry, the crystalline phase, composition, morphology, and pore distribution of HAp–Gel microspheres were all characterized. HAp nanosized crystals and Gel polymers form porous microspheres after blending and exhibit a specific surface area of 158.64 m^2^/g, pore sizes from 3 to 150 nm, and pore volumes of 0.4915 cm^3^/g. These characteristics are suitable for carriers of DOX. Furthermore, by the addition of chitosan during drug loading, its drug-entrapment efficiency increases from 70% to 99% and the release duration increases from a 100% burst within a day to only 45% over half a year since the pores in the composite microspheres provide a shielding effect throughout the degradation period of the chitosan. According to the MTT tests, cell viability of DOX–Chi/HAp–Gel is 57.64% on day 5, similar to the result treated with DOX only. It is concluded that under the protection of pores in the microspheres, the chitosan abundant of hydroxyls combining HAp–Gel and DOX by forming hydrogen bonds indeed enhances the entrapment efficiency, prolongs the releasing period and maintains DOX’s ability to perform medicine functions unaffected after loading.

## 1. Introduction

Hydroxyapatite (HAp) is a major structural ingredient of natural bones and a form of calcium phosphate (CaP) with the chemical formula Ca_10_(PO_4_)_6_(OH)_2_. It has been used clinically in cardiovascular stents and as a carrier for drug-release systems [1] due to its biocompatible, nontoxic, noninflammatory, osteoconductive, and osteointegrative properties [2]. In particular, the porous surface of HAp is advantageous for delivering various pharmaceutical molecules [3,4,5].

A previous study conducted in our laboratory indicated that porous composite microspheres containing HAp and biodegradable gelatin can be prepared successfully by using wet chemical methods [6]. A rat calvarial defect model for bone regeneration capacity was created using fibrin glue, OSTEOSET Bone Graft Substitute and HAp–gelatin composite as scaffolds for comparison. The last one showed the highest osseointegration with 90% of new bone formed in the final stage [6]. It was also reported that a new method involves coating a doxorubicin (DOX)–chitosan–HAp composite on titanium alloy by electrochemical deposition for cancer treatment applications [7].

Cancer has become a major public health problem cause by its high incidence and mortality rates in this century [8]. In Taiwan, cancer ranks as the first among causes of death [9]. Surgery, radiation, chemotherapy, and targeted therapy are all used in cancer treatment. [10]. Chemotherapy is a common treatment for cancer that involves the use of antitumor drugs to retard the growth of tumor cells or damage tumor cells directly. Chemotherapy is considered a systemic therapy for cancer treatment, while sometimes it may also be administered via non-systemic routes. However, it may affect the entire body and kill not only tumor cells but also healthy ones. The side effects of chemotherapy include mouth sores, vomiting, and hair loss [11,12].

The use of targeted drug-delivery systems is seen to have great promise for the treatment of fatal illnesses, particularly cancer and its associated malignant tumors. Particulate carriers, including liposomes, nanoparticles, emulsions, and polymeric micelles for carrying and releasing medications at target areas, are often associated with drug-delivery systems [13,14].

Targeted drug delivery is a direct method of transmitting drugs to patients and increasing the concentration of part of the body phase for some areas of the drug approach. Targeted delivery systems are advantageous because a small dose is required to produce satisfactory results, thereby reducing drug side effects [15].

DOX, also called DOX hydrochloride or adriamycin (commercial), is a powdery, orange-red medicine and its structure contains fat-soluble anthracycline ligand, water-soluble soft red glycosaminoglycan, acidic phenolic hydroxyl and alkaline amino group [16]. DOX is a class I anthracycline antibiotic often used in chemotherapy. This medicine is used for tumor treatment in diseases such as breast cancer, bone cancer, stomach cancer, leukemia, and lung cancer [17]. A total of 40–60 or 20–30 mg of DOX is intravenously injected every 3 weeks or for 3 days, respectively [18]. Because of its non-selective cytotoxicity and dose-dependent congestive heart failure, DOX is an ideal potential drug-delivery candidate [19]. According to the manufacturer instructions, DOX may cause irreversible cardiac toxicity at a total dosage near 550 mg/m^2^ [20], which may lead to congestive heart failure.

Four stages comprise the cell cycle, including the first gap phase (G-1), the synthesis phase (S), the second gap phase (G-2), and the mitotic phase (M). The G-1 phase is the preparation period for cell division. DNA synthesis is used to duplicate genetic material during the S phase. The cytoplasmic materials essential for mitosis and cytokinesis are assembled during the G-2 phase as a result of metabolic changes. The M phase involves cell division. All cell cycle stages, particularly the S phase, are impacted by DOX, through nucleotide intercalation and topoisomerase II inhibition, limiting DNA replication during synthesis [21].

Gelatin is a biocompatible and biodegradable polymer formed through the hydrolysis of collagen. The rate at which antibiotics are released can be reduced by increasing the gelatin membrane’s density [22]. Gelatin is a product of partial collagen hydrolysis, and no differences in amino acid composition are evident between them. The tertiary structure of collagen experiences irreversible denaturation and its chain experiences fragmentation because of the high temperature and harsh pH conditions present throughout the gelatin manufacturing process [23].

Chitosan is a biopolymer of glucosamine and n-acetyl-glucosamine, the most abundant natural polysaccharide containing amino and hydroxyl groups on Earth. It is obtained through the alkaline deacetylation of chitin and widely utilized in environmental and biomedical engineering due to its biocompatibility, biodegradability, hydrophilicity, nontoxicity, and antibacterial properties [24].

Different kinds of biopolymers, for example, particles, membranes, and porous matrices have been used to hybridize with HAp. HAp has been used for various biomedical applications, such as matrices for drug load and release and bone tissue integration materials [25].

In this study, porous HAp microspheres were synthesized in an inorganic aqueous solution containing calcium and phosphate ions by using gelatin as a chelator. The composite microspheres were investigated by using scanning electron microscopy (SEM) for surface morphology, X-ray diffraction (XRD) for crystal structure, porosimetry and chemisorption analysis for specific surface area and pore volume ratio, and inductively coupled plasma mass spectrometry for the ratio of calcium to phosphate. DOX coupled with chitosan loading and simple DOX loading on the HAp–gelatin composite microspheres were aimed at the effects of chitosan on the loading efficiency and releasing duration for their possible applications in drug-delivery systems.

## 2. Materials and Methods

### 2.1. Materials

Calcium nitrate (Ca(NO_3_)_2_∙4H_2_O, 98.5%) and ammonium dihydrogen phosphate (NH_4_H_2_PO_4_, 99%) were purchased from SHOWA. Gelatin powder (bloom 160) was procured from Fluka Chemie, Biochemica 48723 (Buchs, Germany). Chitosan (obtained from crab shells, MW 9.4 × 105 Da and viscosity 372.7 cP) and DOX (D1515) were purchased from Sigma-Aldrich, St. Louis, MO, USA.

### 2.2. Synthesis of Porous HAp–Gelatin Composite Microspheres

A wet chemical method was used to synthesize the HAp–gelatin composite microspheres [26,27]. Deionized water (DI) was used to dissolve NH_4_H_2_PO_4_ and Ca(NO_3_)_2_·4H_2_O to produce calcium and phosphate (CaP) containing solutions, respectively. An aqueous solution containing 4 wt% gelatin was mixed with the CaP solution, and the mixture was agitated for 2 min. To get the precipitate, the final mixture was kept in a water bath at 67 °C for 30 min. The resultant suspension underwent a 10-min centrifugation at 4000 rpm (1789× *g*). The precipitate was gathered and then cleaned with DI water at 67 °C. Finally, to get a white powder made up of the porous HAp–gelatin composite microspheres, filtering (55 mm filter paper, pore volume 5 µm, Advantec) and drying at 50 °C were carried out. In this investigation, several evaluations of the synthesized HAp–gelatin powder were conducted in comparison to the reference HAp powder acquired from Sigma-Aldrich.

### 2.3. Characterization of HAp–Gelatin

#### 2.3.1. SEM

The morphology of the HAp–gelatin and commercial HAp powders after drying were investigated using SEM (JSM-5400, JEOL) and field-emission SEM (FE-SEM; JSM-6700F, JEOL). The components of the powders were analyzed through energy-dispersive X-ray spectrometry (OXFORD INCA ENERGY 400). Before observation, the powders coated with Au were placed on conductive carbon paste affixed to an aluminum mount.

#### 2.3.2. Specific Surface Area Porosimetry and Chemisorption Analysis

A Micromeritics ASAP 2010 nitrogen adsorption equipment was used to calculate the specific surface area of HAp–gelatin microspheres by measuring the nitrogen adsorption of the samples. Through the multipoint Brunauer–Emmett–Teller (BET) technique, adsorption data in the pressure ratio (P/Po) range from 0.02 to 0.45 was used to determine the specific surface area. By employing a desorption isotherm and assuming a cylindrical pore model, the pore size distribution of the produced powders was estimated through the Barret–Joyner–Halender (BJH) technique. The average pore volume and size can be derived from the nitrogen adsorption volume at pressure ratio (P/Po) 0.972.

#### 2.3.3. XRD

XRD (MO3x-HF, Mac Science, Japan) was conducted to detect the crystal diffraction peaks of the powders under Cu Kα radiation (λ = 1.5418 Å) at voltage 40 kV, current 30 mA, 2θ angle range from 10° to 70°, and scanning rate 2°/min. The obtained diffraction patterns conforming to Bragg’s law (2dsinθ = λ) were compared with the Joint Committee on Powder Diffraction Standards (JCPDS)–International Centre for Diffraction Data.

#### 2.3.4. ICP-MS

Measured by ICP–MS (Pe-Sciex Elan 6100 DRC, Wellesley, MA, USA), the Ca and P contents of the powders were ascertained after the powders dissolved in the 1 M HCl solution.

### 2.4. Drug Load and Release Kinetics

#### 2.4.1. DOX Calibration Curves

By using ultraviolet–visible (UV–VIS) spectroscopy (Hitachi U-3010, Tokyo, Japan) to measure the absorbance at 480 nm, DOX calibration curves from 0 to 180 ppm were created.

#### 2.4.2. DOX Coupled with Chitosan Loading and Content Determination

0, 0.125, 0.25, 0.5 and 1 wt% chitosan were dissolved in 0.0033 vol% acetic acid aqueous solutions, then 5 mg DOX powder (0.5 mL, 10 mg/mL) was added in, respectively, and assigned to drug loading (DL) 1, 2, 3, 4 and 5. After mixing for 20 min, a total of 25 mg of HAp–gelation powder was mixed in and blended at 80 rpm (0.716× *g*) for 48 h. To measure the residual drug, the DOX-loaded particles were dried at 60 °C for 48 h, the dried DOX-HAp was then removed, and 1 mL of DI water was added to the vial to dissolve the remaining drug. The concentration of DOX in the supernatant was estimated by the calibration curve. By deducting the quantity of DOX in the supernatant from the initial amount of DOX, the DOX encapsulated in the HAp–gelatin powders was estimated. Then the drug-loading (DL) content and drug-entrapment efficiency (EE) of DOX were calculated, respectively, by the following equations [6]:EE (%) = (initial DOX amount − DOX amount in the supernatant) × 100/initial DOX amount,(1)
DL (%) = (initial DOX amount − DOX amount in the supernatant) × 100/microsphere amount,(2)

#### 2.4.3. In Vitro DOX Releasing from DOX-Loaded Composite Microspheres

In vitro release experiments were conducted by suspending 25 mg of DOX-loaded microspheres in 10 mL phosphate buffered solution (PBS; Sigma-Aldrich) in vials. The vials were placed in an orbital shaking bath at 37 °C and rotated at 80 rpm (0.716× *g*). At each designated time point, 1-mL aliquots of the release medium were removed and immediately replaced with fresh PBS, all carried out in triplicate. The amount of DOX release was calculated by comparing the drug concentration before and after release using UV–VIS spectroscopy and the calibration curve. The adjusted mass of the released DOX was derived by the following equation:(3)$$M_{c}=M_{t}+\frac{v}{V}\sum_{0}^{t-1}M_{t}$$

M*_c_* is the corrected mass at time *t*, M*_t_* the apparent mass at time *t*, *v* the taken volume from the sample, and *V* the total volume of the release fluids.

### 2.5. Cell Experiment

#### 2.5.1. Cell Culture

Human osteosarcoma-derived G-292 cells (ATCC CRL-1423) were cultured in McCoy’s 5A (modified) medium with 10% FBS in a 5% CO_2_ atmosphere at 37 °C. The media was renewed with a fresh one every three days up to the time when the cells attained confluence. After being separated with 0.25% trypsin-EDTA solution (GIBCO, Cana-da), the cells were seeded into T75 flask cell culture plates with 1 × 10^6^ cells per flask and fresh media.

#### 2.5.2. Medium Extracted from Immersion Test

HAp–gelatin and DOX–chitosan-loaded HAp–gelatin powders were immersed in 10% fetal bovine serum (FBS; Biological Industries, Beit HaEmek, Israel) containing McCoy’s 5A (modified) medium at 37 °C for 6 h. After the immersion test, the medium was extracted for the cytotoxicity test. The amount of DOX powder was tuned to the same as that of the DOX-chitosan-loaded HAp–gelatin porous composite microspheres, for checking the medical effectiveness of DOX after loading.

#### 2.5.3. Cytotoxicity Tests

An in vitro cytotoxicity test was performed to estimate the biocompatibility of the synthesized HAp–gelatin, DOX–chitosan-loaded HAp–gelatin porous composite microspheres and pure DOX powder (killed group). The indirect contact test, in compliance with ISO10993-5 [28], was conducted to evaluate in vitro cytotoxicity. In all, 10^4^ cells/100 μL of the medium were seeded into the individual well of a 96-well plate, and the cells were then left to attach for 24 h. Following that, the medium extracted from the immersion test was added in and then incubated for 1, 3 and 5 days, and the medium was refreshed once after day 3. After incubating, the 3-(4,5-dimethylthiazol-2-yl)-2,5-diphenyl-2H-tetrazolium bromide (MTT) assay was used to determine cell viability. The tests were carried out in triplicate.

#### 2.5.4. MTT Assay

To measure mitochondrial reductive function, the MTT colorimetric assay is often employed. It is a suitable indicator of cell death or inhibited growth.

The culture medium was taken out of several well groups to quantify MTT activity after 1, 3, and 5 days. Each well was then filled with 50 µL of MTT solution, created by dissolving 5 mg thiazolyl blue tetrazolium bromide (SIGMA M5655) in 1 mL PBS. The solution was then left to incubate for two hours. After removing the media, the formazan crystals were then dissolved in dimethyl sulfoxide (Merck, Darmstadt, Germany) 100 µL. In a gentle rocking motion for 30 min, the culture plates were solubilized. An ELISA reader (Stat Fax-2100, Awareness Technology, Inc., Palm City, FL, USA) was then used to measure the optical intensity at 545 nm. The tests were carried out in triplicate.

## 3. Results and Discussion

### 3.1. Characterization of HAp–Gelatin Porous Composite Microspheres

FE-SEM images of the HAp–gelatin porous composite microspheres and the commercial HAp powder purchased from Sigma-Aldrich are presented in Figure 1a,b. The HAp–gelatin porous composite microspheres comprised numerous pores within petal-like flakes. The average size of the porous HAp–gelation composite microspheres was approximately 45 μm. The commercial HAp powder exhibited a nonporous morphology. Anticancer drugs can be loaded onto the synthesized composite microspheres because they have a homogeneous and spherical shape as well as high porosity. Thus, the synthesized powders may have high application potential in the biomedical field.

The Ca and P contents of HAp–gelation porous composite determined through ICP–MS are listed in Table 1. The derived Ca/P ratio of the commercial HAp was 1.57, and that of the HAp–gelatin porous composite microspheres was 1.52, while the normally ratio should be 1.67. Thus, the synthesized microspheres comprised HAp with a relatively low calcium content, also named the calcium deficient HAp.

#### 3.1.1. Specific Surface Area and Pore Volume of HAp–Gelatin Composite Microspheres

The BET nitrogen adsorption–desorption isotherm of the synthesized microspheres is presented in Figure 2a. Typical hysteresis loops are usually observed for multilayer adsorption. Four typical adsorption curves exist as depicted in a previous report [29]. The H1 curve occurs when agglomerated uniform spheres form a porous material. The H2 curve is not clearly defined because numerous effects must be considered in its analysis. The H3 curve occurs when the porous material contains microflakes. The H4 hysteresis loop is formed when the pores are narrow. The hysteresis loop of the prepared microspheres shown in Figure 2a is similar to H3, consistent with the petal-like flakes observed by FE-SEM shown in Figure 1a.

According to BET theoretical analysis, the obtained specific surface area of the microspheres is 158.64 m^2^/g between BJH adsorption (123.92 m^2^/g) and BJH desorption (172.97 m^2^/g), which are derived from the specific surface area of pores sized from 17 to 3000 Å, as listed in Table 2. The pore volume is 0.4915 cm^3^/g, and the porosity is 56%, obtained by the BJH adsorption and desorption method. As displayed in Figure 2b, the BJH desorption pore size ranged from 30 to 1500 Å. Volume distribution peaks were observed at 20 (the smaller) and 660 Å (the greater), indicating the high porosity and surface area of the microspheres. These properties allow the synthesized materials to have strong drug-carrier potential.

#### 3.1.2. XRD Patterns

XRD patterns of the prepared composite microspheres and commercial HAp are depicted in Figure 3. Compared with the commercial HAp, the prepared HAp–gelatin porous composite microspheres exhibited fewer sharp peaks, corresponding to the main X-ray diffraction peaks such as crstal planes (002), (211), (112), (202), (222) and (213) of HAp identified by JCPDS 9-432. The HAp–gelatin porous composite microspheres also exhibit the broaden peaks with relatively low intensity, indicating a poor crystallinity and/or nanosized crystal.

### 3.2. Drug Loading and Releasing

#### 3.2.1. UV–VIS Spectrum for DOX Concentrations

The UV–VIS spectrum of DOX is presented in Figure 4a. Characteristic absorption peaks were observed at 252, 291, and 480 nm. The absorbance at 480 nm was selected to identify the DOX content. The calibration curve of the absorbance at 480 nm versus DOX concentration (ppm) in DI water is shown in Figure 4b. It follows the linear equation y = 9.0086x − 0.0067, and R^2^ = 0.9961 for the DOX concentration under 180 ppm.

#### 3.2.2. Mechanisms of DOX Loading and Releasing

DOX-loaded HAp composite microspheres were prepared in chitosan (Chi) solution at 37 °C for 48 h. A control sample containing only DOX and composite microspheres without chitosan powder was used to illustrate the effect of chitosan. In addition, the weight ratios of DOX to the HAp–gelatin porous composite microspheres to chitosan are 1:5:0, 1:5:0.125, 1:5:0.25, 1:5:0.5 and 1:5:1 corresponding to DL1, DL2, DL3, DL4 and DL5. The chemical structures of DOX and chitosan and the loading procedure are depicted in Figure 5. During loading, DOX was first bonded with chitosan through hydrogen bonds to produce DOX–Chi; this argument is also supported by the previous report [30]. Then HAp–gelatin was added to form DOX–Chi/HAp–gelatin, which was subsequently dried in a furnace at 50°C for two days. The surface morphology of the HAp–gelatin porous composite microspheres after drug loading for DL1, DL2, DL3, DL4, and DL5 shows that the spherical shape and pores gradually disappear when the content of chitosan increases as seen in Figure 6, meaning that the DOX–Chi fully fills the pores and cover the surface of microspheres. The DL and EE with various chitosan weights are listed in Table 3. EE and DL increase when the weight of the chitosan is increased due to the extraordinary hydrogen bonding between chitosan and DOX.

The drug-release profiles of various formulations are presented in Figure 7. The drug-loaded HAp–gelatin porous composite microspheres without chitosan exhibited 100% drug burst for DL1 and DOX–Chi. The formulations added with chitosan exhibited a drug release of only 21% after 24 h for DL5 as shown in Figure 7a. The DOX on the HAp–gelatin microspheres was easily dissolved by PBS, indicating that the bonding strength between DOX and HAp–gelatin microspheres is weak. Figure 7b indicates that the burst drug concentration increases when the chitosan weight decreases for DL2, DL3, DL4 and DL5. The drug release is only 45% after 24 weeks for DL5 as shown in Figure 8, including five steps of in vitro drug release. The detailed drug-release process and the related mechanisms are illustrated in Figure 9. The first step is the initial drug burst in 2 h, ascribed to the dissolution of drug powders dried on the surface of particle (Figure 9a). The second step from hour 3 to hour 24 exhibits a moderate release rate, due to the debonding from the drug to the surface of chitosan (Figure 9b). The third step from day 2 to day 6 is the drug concentration in the transition equilibrium state between the solution and the drug carrier illustrated in Figure 9c. The surface of the DOX/HAp–gelatin is still covered by chitosan, which is also revealed by the right SEM image. After the 1 mL medium was sucked out by pipette and refreshed by PBS 14 times in a week, the drug concentration decreased to a sufficiently low level for processing the next step. The fourth step from week 2 to week 4 presents a greater drug releasing rate, resulting from the swelling and degradation of chitosan on the surface of microsphere as illustrated in Figure 9d, similar to the previous report that the obvious degradation of chitosan occurred after one week and consistent with the right SEM image, where several pores show up after the gradual degradation of chitosan [31]. The final step from week 5 to week 24 reveals a gentle release rate, caused by the drug release from the macropores and mesopores of the microspheres, as illustrated in Figure 9e, and more pores are found on the surface of DOX/HAp–gelatin after more degradation of chitosan in pores, as shown in the right SEM image. The release curve of step 5 (from week 5) can be explained by y% = 0.6384x (week)% + 29.221% which shows the drug is still in the process of release, and the presence of enzymes in the human body will accelerate the degradation of chitosan [32]; therefore, the drug in DOX/HAp–gelatin will release faster in vivo. Beside forming more hydrogen bonds among HAp, DOX and chitosan than those between HAp and DOX, the pores in the microsphere composite provide a shielding effect for the inserted DOX–Chi from the degradation, leading to the slower releasing rate and exhibiting only 45% drug release after 24 weeks.

### 3.3. MTT Test

Figure 10 presents the results of the MTT test of the G-292 cells in the control, HAp–gelatin, DOX, and DOX/chitosan/HAp–gelatin (DL5) groups. The number of surviving cells in the well containing the DOX-loaded microspheres was considerably lower than that in the unloaded one. The cell viability of DL5 is only 89.05%, 63.96%, and 57.64% of the control for tests on day 1, day 3, and day 5, respectively, similar to the results of DOX, indicating that DOX ability of DL5 to perform medicine functions is unaffected during processing the drug loading. Moreover, the cell viability of HAp–gelatin is much greater than the control on days 3 and 5. The ability of accelerating the cell growth of the prepared porous microspheres might be due to the ion release of calcium and phosphate in the extracted medium, which was found in the previous report [33].

## 4. Conclusions

The HAp–gelatin composite microspheres composed of bioactive HAp and gelatin are well prepared through wet chemical methods. The obtained microspheres sized around 45 μm are porous, with a mesopore volume of 0.49 cm^3^/g, and a high specific surface area of 158.64 m^2^/g. Thus, the prepared microspheres exhibit strong potential as drug carriers. However, the weak bonding strength between DOX and HAp–gelatin resulted in a 100% drug burst on the first day. The bonding strength is enhanced by the addition of chitosan, which strongly combines DOX and inserts in pores of the HAp–gelatin composite microspheres due to its abundant hydroxyls, finally leading to the drug-entrapment efficiency being enhanced from 70% to 99% and the release period being extended to more than half a year. Five steps of in vitro drug release are observed. The first step involving initial drug burst in 2 h is caused by the dissolution of dried drug powders. The second step from hour 3 to hour 24 exhibits a medium release rate, resulting from the debonding of the drug from the chitosan surface. The third step from day 2 to day 6 is the transition equilibrium state of the drug concentration between the solution and the drug carrier. After the removals of the medium and 14 replacements with new PBS, the drug concentration decreases to a sufficiently low level. The fourth step from week 2 to week 4 reveals a greater drug releasing rate, due to the swelling of chitosan on the surface of microspheres. The final step, occurring from week 5, exhibits a gentle release rate, ascribed to the drug debonding from the chitosan inserted in the macropores and mesopores of the microspheres. Under the protection of the pores in the microspheres, the chitosan abundant of hydroxyls combining HAp–gelatin and DOX by forming hydrogen bonds are speculated to enhance entrapment efficiency, extend the drug-release period, and retain the medicine efficacy of DOX after being loaded.

## Figures and Tables

**Figure 1 polymers-14-04276-f001:**
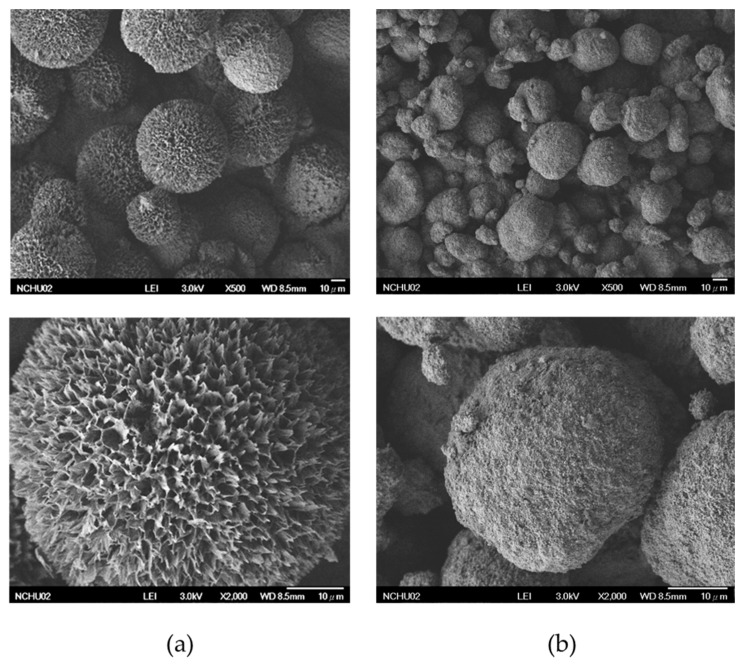
FE-SEM micrographs of (**a**) the synthesized HAp–gelatin porous composite microspheres and (**b**) the commercial HAp from Sigma-Aldrich.

**Figure 2 polymers-14-04276-f002:**
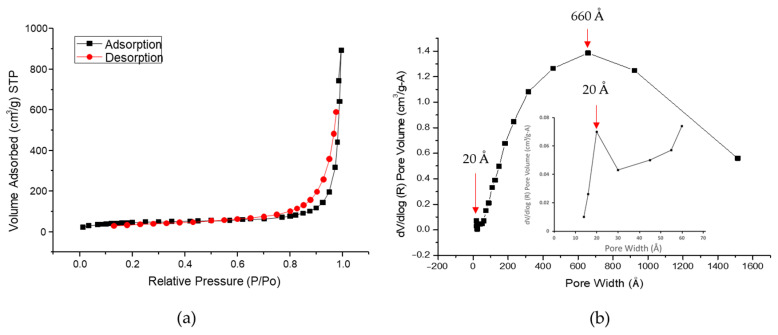
(**a**) BET nitrogen adsorption–desorption isotherm and (**b**) distribution of BJH desorption pore size and the enlarged version around 20 Å is also inserted.

**Figure 3 polymers-14-04276-f003:**
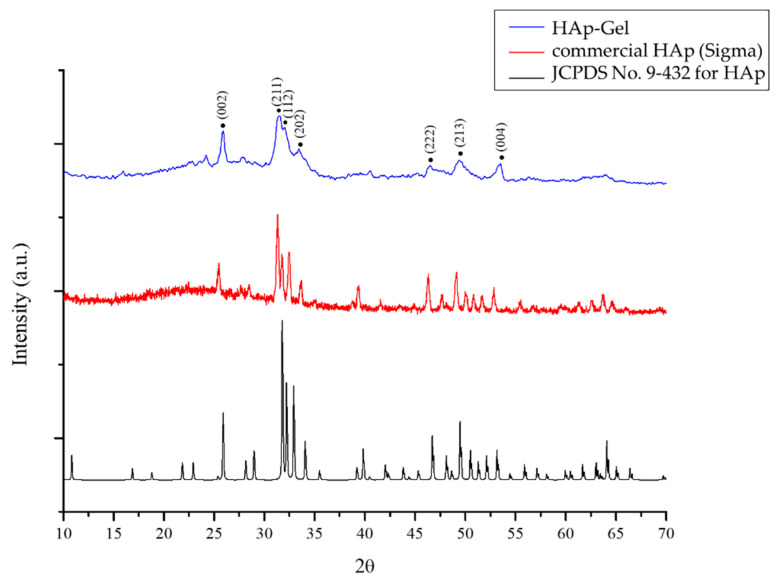
XRD patterns of the HAp–gelatin porous composite microspheres, the commercial HAp, and JCPDS file No. 9-432 for HAp.

**Figure 4 polymers-14-04276-f004:**
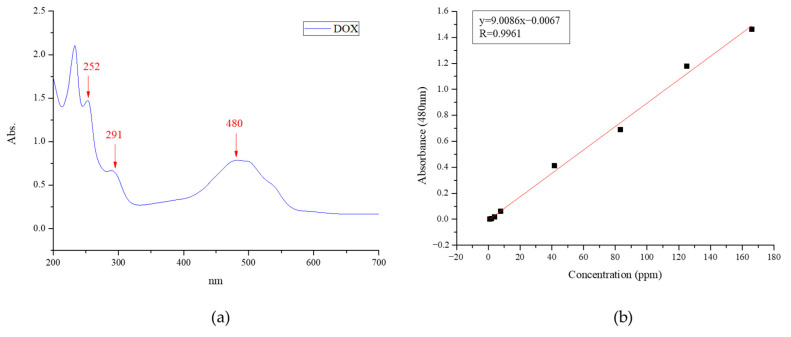
(**a**) UV–VIS spectrum of the DOX solution. (**b**) Calibration curve of absorbance at 480 nm versus the DOX concentration (ppm) in DI water.

**Figure 5 polymers-14-04276-f005:**
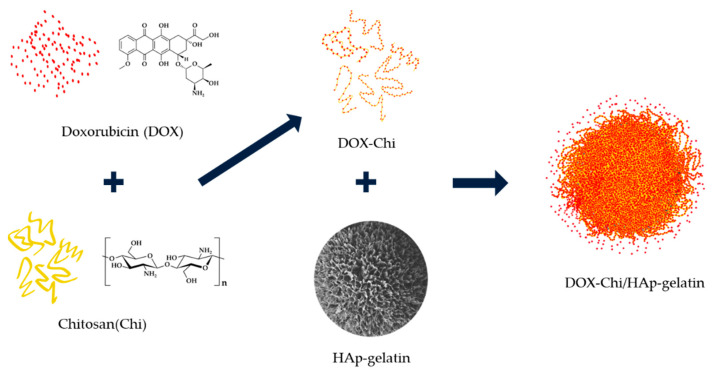
The chemical structures of DOX and chitosan are drawn and presented by red dot and yellow fiber, respectively. They can form hydrogen bonds to combine with each other due to abundant hydroxyls, then insert into the pores of HAp–gelatin through blending, and finally form the DOX–Chi/HAp–gelatin composite.

**Figure 6 polymers-14-04276-f006:**
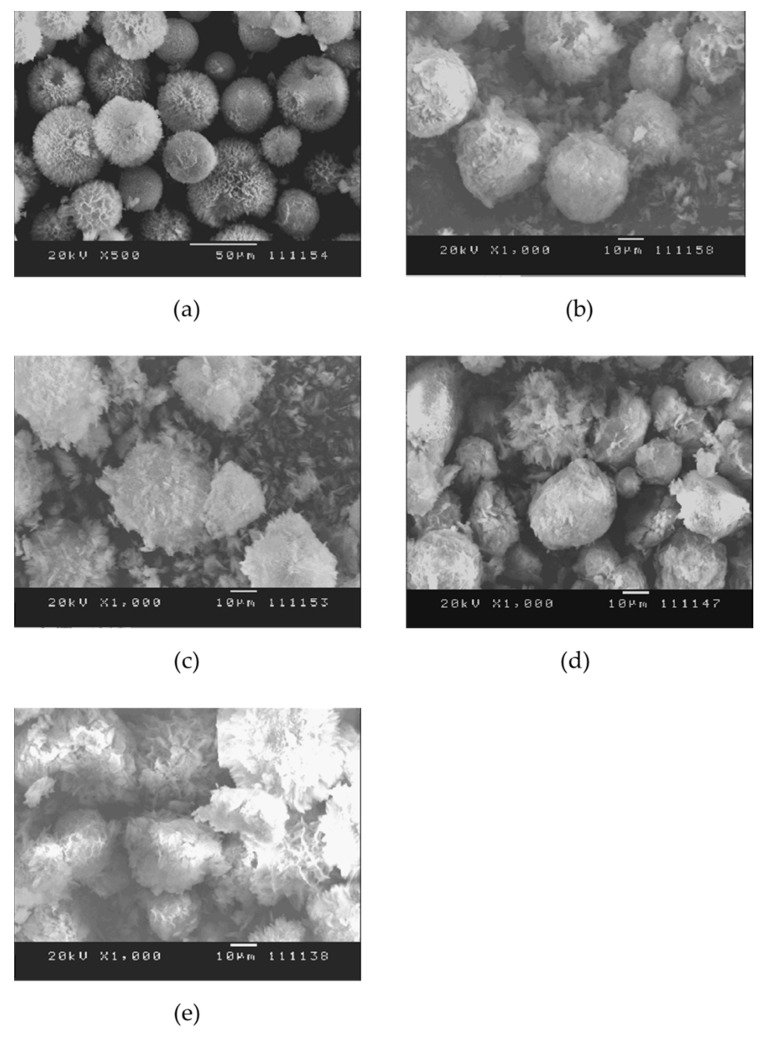
The surface morphology of the HAp–gelatin porous composite microspheres after drug loading for (**a**) DL1, (**b**) DL2, (**c**) DL3, (**d**) DL4, and (**e**) DL5.

**Figure 7 polymers-14-04276-f007:**
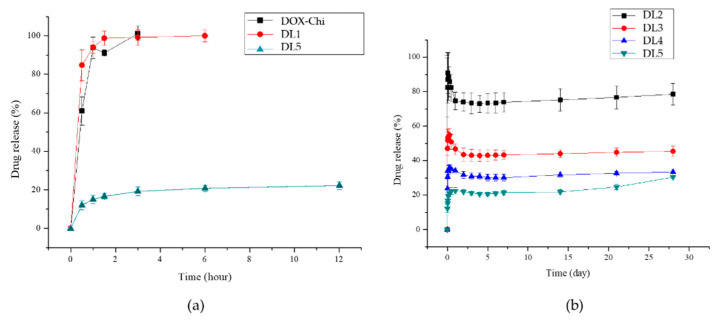
(**a**) DOX release profiles of DOX/chitosan, DL1, and DL5 samples over 24 h. (**b**) DOX release profiles of DL2, DL3, DL4 and DL5 over 1 month.

**Figure 8 polymers-14-04276-f008:**
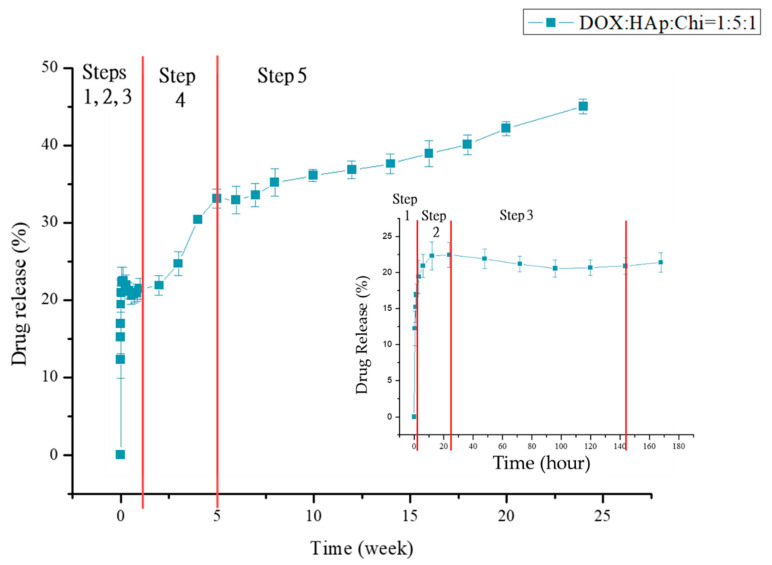
Drug-release profiles of DL5 for 24 weeks present five steps and the right inserted profile is the enlargement of the initial three steps.

**Figure 9 polymers-14-04276-f009:**
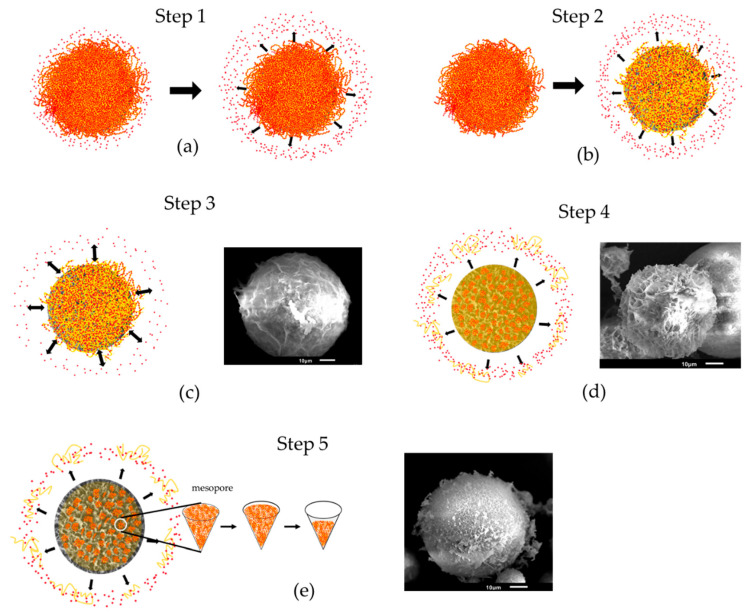
In vitro drug-release process illustrates five steps related to Figure 8, including (**a**) the dissolution of dried drug powders on the surface in step 1, (**b**) drug deboned from the surface of chitosan in step 2, (**c**) drug concentration of transition equilibrium state between solution and drug carrier in step 3, (**d**) the swelling and degradation of chitosan on the surface of microspheres in step 4, (**e**) the swelling and degradation of chitosan in mesopores and/or macropores of microspheres in step 5. SEM images are also shown in steps 3, 4 and 5.

**Figure 10 polymers-14-04276-f010:**
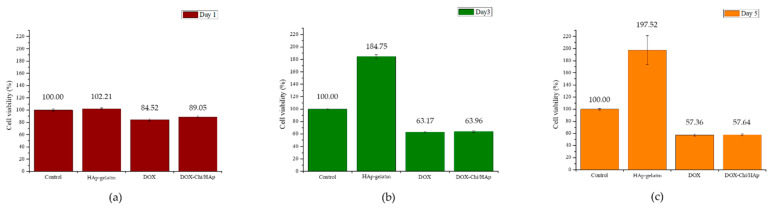
G-292 cell viability of the control, HAp–gelatin, DOX, and DOX/chitosan/HAp–gelatin samples after (**a**) 1, (**b**) 3, and (**c**) 5 days.

**Table 1 polymers-14-04276-t001:** Ca content, P content, and Ca/P atomic ratio of the commercial HAp and the prepared HAp–gelatin microspheres obtained from ICP–MS analyses.

	Ca (wt%)	Ca (mol)	P (wt%)	P (mol)	Ca/P
Commercial HAp (sigma)	37.7	0.95	18.6	0.6	1.57
HAp–gelatin microsphere	28.1	0.7	14.3	0.46	1.52

**Table 2 polymers-14-04276-t002:** BET and BJH analyses of specific surface area, pore volume, and average pore size for the prepared HAp–gelatin microspheres.

	BET Adsorption Theory	BJH Adsorption	BJH Desorption
Specific surface area	158.64 m^2^/g	123.92 m^2^/g	172.97 m^2^/g
Pore volume		0.4915 cm^3^/g	
Average pore size	123.93 Å	442.58 Å	319.24 Å

**Table 3 polymers-14-04276-t003:** DL and EE with various weight ratios of HAp–gelatin porous composite microspheres to chitosan for DL1, DL2, DL3, DL4 and DL5.

	DOX (mg)	HAp–Gel (mg)	Chitosan (mg)	DL (%)	EE (%)
DL1	5	25	0	13.95 ± 0.29	69.76 ± 5.81
DL2	5	25	0.625	13.79 ± 0.53	68.93 ± 10.69
DL3	5	25	1.25	17.53 ± 0.17	87.67 ± 3.44
DL4	5	25	2.5	19.55 ± 0.06	97.75 ± 1.28
DL5	5	25	5	19.88 ± 0.01	99.39 ± 0.29

## Data Availability

Not applicable.

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
