# Peer review of "Effects of Chitosan on Loading and Releasing for Doxorubicin Loaded Porous Hydroxyapatite–Gelatin Composite Microspheres"

_polymers, 2022, doi:10.3390/polym14204276_

Round 1
Reviewer 1 Report
Comments:
The manuscript “Effects of chitosan on loading and releasing for doxorubicin loaded porous hydroxyapatite-gelatin composite microspheres” by Shiow-Kang Yen and colleagues introduced DOX loaded HAp-gelatin composite microspheres incorporating with chitosan. The microsphere had high drug entrapment efficiency and extended the release period to more than half a year. This work is highly interesting and has good potential in the drug delivery field. This manuscript is also well written. However, the reviewer believes that additional points of clarifications could potentially be addressed to further strengthen the manuscript.
1. In line 212, there were double space between “Awareness Technology, INC.) and was then used to …” please correct this.
2. In Figure 2b, it is hard to see the peak of 20A peak, I would recommend changing the x-axis scale to show this bimodal pore size distribution clearer.
3. In line 287, the author state that the DOX loaded due to the hydrogen bonds. The author should provide some experiments to show the drug loading due to the hydrogen bonding.
4. In Figure 7, DL5 line is green color, but in Figure 8, it becomes to blue color which is the same color as DL4 in Figure 7. I would recommend keeping this consistent.
5. In Figure 8, the author stated the 45% of DOX released from DL5 after 24 weeks, which means there are still 55% of DOX inside the DL5 group. For me, it seems the chitosan hold the drug and DOX will never release out. From week 5 to week 24, there was only 10-15% of DOX released. Does such little release percentage of DOX have therapeutic effects? If not, then the chitosan-based microsphere won’t be ideal for in vivo.
6. Drug release typically related to the degradation of carriers. I would recommend the author include the degradation test of the DL5 group.
7. In Figure 10, the cell viability assay was performed in 1, 3 and 5 days. Typically, the author using the 96 well plate to do this experiment. You treated the cells with your samples and wait for 1, 3 or 5 days. Typically, the cells will consume all the nutrients and the cell media will become yellow after 5 days, this will also cause some cell toxicity. How does the author explain this?
8. In Figure 10 b and c, the HAp-gelatin groups outperform to 184% and 197.52%, this cannot reveal the excellent biocompatibility. This can only be explained the HAp gelatin groups accelerate the cell growth, which doesn’t make sense to me. It might be your MTT assay is not working properly. The author should include a killed group (negative control) where some reagent to kills to cells to prove that the MTT assay is working properly.
Author Response
The authors are grateful for the comments to improve the quality of submitted manuscript.

Reviewer 2 Report
General comments
The work presented by Wu, et al focuses on the development of hydroxyapatite-gelatin microspheres for doxorubicin loading and cancer treatment. Starting with a comparison between the in-house developed microparticles and commercial ones, the authors further functionalize the system with different amounts of chitosan and doxorubicin. Overall, the manuscript is well written and presents very interesting results. English should be revised as some sentences lack meaning. Some minor aspects require further adjustments before being considered for publication in Polymers.
Additional comments
L48-49 Although less frequently, chemotherapy may also be administered via non-systemic routes.
L95-L99 This information is not relevant for the manuscript, as it is not related to the experimental approach used by the authors.
L113 Include type of chitosan (MW, etc )
L122 Use g instead of rpm
L123 Include filter type and pore size
L151 Include HCL molarity
L153 Include calibration curve range
L194 Include the n of the experiments
L228-229 What implications can be inferred from this sentence?
L264-266 What implications can be inferred from this sentence?
L285-286 Figure 5 does not include any chemical structures.
L337 According to the text, phase III lasts until day 6 (144h), with phase IV starting after the end of the first week. Correct the vertical bar of the graphs, accordingly.
L357 Include cell viability values over the bars to improve readability.
English/Grammar
Correct the following sentences:
L67 Correct to “non-selective cytotoxicity”
L20-21 “Cell viability of DOX-Chi/Hap-Gel is”
L105 “DOX coupled with chitosan and without chitosan loadings”
L166-L163 Replace DLC and DEE by DL (drug loading) and EE (entrapment efficiency) respectively.
L168, L280 Use “DOX release” or “drug release” instead
L193 Replace “Dox after loaded” by “Dox after loading”
L294-295 The sentence does not make sense
L306, L308 – Replace “mixtures” by “formulations” or equivalent
L311-313 The sentence does not make sense
Author Response
The authors are grateful for the comments to upgrade the quality of submitted manuscript.

Round 2
Reviewer 1 Report
The authors have addressed all my comments, so I will recommend to accept this manuscript for MDPI-polymers.